# Ischemia Reperfusion Injury: Mechanisms of Damage/Protection and Novel Strategies for Cardiac Recovery/Regeneration

**DOI:** 10.3390/ijms20205024

**Published:** 2019-10-11

**Authors:** Andrea Caccioppo, Luca Franchin, Alberto Grosso, Filippo Angelini, Fabrizio D’Ascenzo, Maria Felice Brizzi

**Affiliations:** 1Department of Medical Sciences, University of Turin, 10124 Torino, Italy; andrea.caccioppo@gmail.com (A.C.); alberto.grosso@edu.unito.it (A.G.); 2Division of Cardiology, Department of Medical Sciences, University of Turin, 10124 Torino, Italy; luca.franchin@gmail.com (L.F.); filippoangelini90@gmail.com (F.A.)

**Keywords:** cardiac ischemic disease, cardiac regeneration, stem cells, exosomes, therapeutic approaches

## Abstract

Ischemic diseases in an aging population pose a heavy social encumbrance. Moreover, current therapeutic approaches, which aimed to prevent or minimize ischemia-induced damage, are associated with relevant costs for healthcare systems. Early reperfusion by primary percutaneous coronary intervention (PPCI) has undoubtedly improved patient’s outcomes; however, the prevention of long-term complications is still an unmet need. To face these hurdles and improve patient’s outcomes, novel pharmacological and interventional approaches, alone or in combination, reducing myocardium oxygen consumption or supplying blood flow via collateral vessels have been proposed. A number of clinical trials are ongoing to validate their efficacy on patient’s outcomes. Alternative options, including stem cell-based therapies, have been evaluated to improve cardiac regeneration and prevent scar formation. However, due to the lack of long-term engraftment, more recently, great attention has been devoted to their paracrine mediators, including exosomes (Exo) and microvesicles (MV). Indeed, Exo and MV are both currently considered to be one of the most promising therapeutic strategies in regenerative medicine. As a matter of fact, MV and Exo that are released from stem cells of different origin have been evaluated for their healing properties in ischemia reperfusion (I/R) settings. Therefore, this review will first summarize mechanisms of cardiac damage and protection after I/R damage to track the paths through which more appropriate interventional and/or molecular-based targeted therapies should be addressed. Moreover, it will provide insights on novel non-invasive/invasive interventional strategies and on Exo-based therapies as a challenge for improving patient’s long-term complications. Finally, approaches for improving Exo healing properties, and topics still unsolved to move towards Exo clinical application will be discussed.

## 1. Introduction

The ischemic cascade was first described over 30 years ago [1]. The imbalance between myocardial oxygen supply and demand translates into angina and myocardium necrosis if not promptly recognized and treated in an acute setting. As a result, necrosis turns into fibrosis, which causes a reduction of both myocardial contraction and left ventricular ejection fraction (LVEF), and eventually to heart failure (HF). Several efforts have been devoted to reduce long-term mortality in patients with HF progression upon myocardial injury [2]. However, after primary percutaneous coronary intervention (PPCI) and optimization of medical therapies, no further improvements in clinical outcomes have been achieved. This has spurred the scientific community to search for alternative therapeutic options, which range from pharmacological/interventional approaches to cell-based therapy [3]. Novel interventional protocols have been proposed to reduce myocardial oxygen consumption and/or to improve heart collateral blood supply. Moreover, clinical trials exploiting multitarget therapeutic options are ongoing and they could represent a future challenge [4]. More recently, different cell sources and their derivatives, including extracellular vesicles (EV), have been also investigated for their potential application [5,6].

EV are small vesicles with a lipid bilayer membrane that is secreted by almost all cell types. EV are considered a novel paracrine and endocrine mechanism of cell-to-cell communication. The transfer of their composite cargo, consisting of lipids, proteins, RNA, including mRNA [7], microRNA (miR) [8], and long noncoding (lnc)RNA, drives their functional effects on target cells [9,10]. Exocarta (see at www.exocarta.org) [11] or EVpedia (http://www.evpedia.org/) [12] provide the updated list of known molecules that are carried by EV.

EV were first described as plasma membrane fragments released by platelets as part of the coagulation process by Peter Wolf in 1967 [13]. From then, a number of studies have provided evidence for their role in many biological processes, including inflammation, angiogenesis, and coagulation [14,15,16,17]. More recently, the therapeutic potential of EV has been demonstrated in different clinical settings [15,16,18,19]. Recently, their potential role in myocardial ischemia/reperfusion (I/R) injury has been evaluated [20,21], which provides new insights that are exploitable for therapeutic purposes. Indeed, EV have been proposed as a novel option to interfere with or prevent scar formation in myocardial infarction (MI) [22,23,24,25,26].

The first part of this review will introduce the most relevant mechanisms of damage and cardioprotection, as the therapeutic strategies are mainly based on specific targeting approaches. Moreover, it will provide an overview on multitarget pharmacological/non-pharmacological approaches and on the promising mechanical reperfusion techniques that were developed to protect myocardium from I/R damage. In addition, recent advances in using EV for cardioprotection/cardiac regeneration will be reported. Finally, approaches for improving EV healing properties and the hurdles still unsolved for moving from bench to bedside will be discussed.

## 2. Reperfusion Injury

MI results from both coronary artery occlusion and reperfusion damage. The reperfusion time, the presence of collateral vessels and the patient’s hemodynamic status dictate the fate of the myocardial area at risk. However, despite advances in PPCI, the management of microvascular damage in the reperfused myocardium remains a challenge. As originally demonstrated by Maroko et al. [27] and further supported by a number of evidences [28,29,30], reperfusion strategies are crucial for myocardial salvage. In current clinical practice reperfusion, when appropriate, is the cornerstone in the management of acute coronary syndrome (ACS) [30,31]. The sooner blood flow will be restored, better outcomes will be obtained, particularly regarding the overall mortality [32]. A brief description will be provided, as several mechanisms contribute to myocardial salvage upon reperfusion.

### 2.1. Lethal Reperfusion Injury

The questioned concept of lethal reperfusion injury has been currently accepted, owing to the results that were obtained by Zhao et al. [33]. The authors proved that a reduction of the infarct size occurred in dogs that were subjected to post-conditioning strategies. In particular, they have shown that a progressive reperfusion could be obtained by intermittent inflation of a balloon inside the infarct related artery, just after blood reflow. Such a protocol clearly demonstrated that a reduction in the infarct size was independent of the ischemic time, while being dependent on reperfusion itself.

### 2.2. Microvasculature Damage

Microvascular damage, also known as no-reflow phenomenon, was first described by Kloner et al. [34], as a crucial determinant of myocardial injury. This phenomenon occurs in 10% to 30% of ST elevation MI (STEMI) patients after reperfusion [35,36] and it is considered to be a negative prognostic factor [37]. A number of hypotheses have been postulated for explaining the no-reflow phenomenon: the release of cellular debris, the presence of platelet and leucocyte aggregates, the vasospasm induced by vasoconstrictors released after reperfusion [38,39], myocardium edema [40], and direct capillary destruction [34].

### 2.3. Mitochondrial Permeability Transition Pore (mPTP)

Independent of the mechanisms accounting for the reperfusion damage, mPTP opening is considered to be the key driver of injury [41]. Briefly, mPTP is formed by a ring of the *c* subunits of the adenosine triphosphate (ATP) synthase [42]. During ischemia, the metabolic shift towards anaerobiosis promotes calcium overload and long-fatty acid accumulation, which would lead to mPTP opening. However, the simultaneous reduction of cytosolic pH prevents this event [43]. When reperfusion occurs, a sudden O2 influx, together with the production of reactive oxygen species (ROS) and pH neutralization, promote mPTP opening [41,44]. This translates in ATP depletion and the loss of ionic homeostasis [45].

mPTP opening drives cell towards necrosis (if stably opened) or apoptosis (if transiently opened), depending on the release of cytochrome *c* or the activation of caspase-9 and 3 receptively [41]. The biological relevance of the mPTP functional state was demonstrated in preconditioning setting while using the mPTP inhibitor, cyclosporine A [46]. Indeed, it was found that cyclosporine A, by binding to its receptor, the mitochondrial cyclophilin D, was able to reduce the infarct size in MI preclinical models [47]. However, both cyclosporine A [48] and TRO40303 [49] (a different mPTP inhibitor), administered in two different clinical trials, failed to demonstrate any clinical benefits. This suggests that a deep knowledge of the complex cascade controlling mPTP opening is still missing.

## 3. Cardioprotective Pathways

Apoptotic cell death is the final event occurring after I/R injury [50]. Therefore, interfering with or preventing apoptotic cell death would prevent MI-associated damage and impact patient’s clinical outcomes. A large number of receptors or intracellular signaling pathways that are involved in cardioprotection have been described. Herein, a brief summary of the most relevant pathways will be reported.

### 3.1. The Anti-Apoptotic Pro-Surviving Pathway: The “Reperfusion Injury Salvage Kinase” (RISK) Pathway 

The RISK pathway, which encompasses the activation of the phosphoinositide 3-kinase (PI3K)-AKT and Mitogen-Activated protein kinase (MEK)/Extracellular Signal-Regulated Kinase (ERK) cascade, was first described by Yellon et al. [51]. However, from the original report, a number of effectors acting on AKT [52,53] have been described, as to promote protection or damage, depending on their acute or long-lasting activation [54]. In preclinical models of cardiac I/R, the activation of the RISK kinase cascade occurs during both preconditioning cycles and early reperfusion [55], and it modulates the mPTP functional state, by converging on the Glycogen Synthase Kinase-3β (GSK-3β) [56,57,58]. Moreover, it has been shown that the overexpression of Phosphatase and Tensin Homolog (PTEN), which controls the PI3K-AKT kinase activity [59], prevents protection driven by conditioning strategies [60], while its suppression rescues cardioprotection [53,61]. Therefore, PTEN has been proposed as a potential I/R target [62,63].

### 3.2. The Survivor Activating Factor Enhancement (SAFE) Pathway 

The SAFE pathway is an alternative survival kinase cascade that converges on the Signal Transducer and Activator of Transcription-3 (STAT3). In mice, the inhibition of the mPTP opening [64] and the expression of the mitochondrial protein Optical Atrophy-1 (OPA1) [65] are under the control of STAT3 tyrosine and serine phosphorylation [66,67], and they represent the most relevant mechanisms of STAT3-mediated cardioprotection. Tumor Necrosis Factor Receptor-2 (TNFR2), via Janus Kinase (JAK), is the most relevant mediator of STAT3 activation in cardiomyocyte [68,69,70,71,72]. Moreover, it has been reported that the activation of ERK [40,41] and the inhibition of Forkhead Box O-1 (FOXO-1) [69] contribute to the cross-talk between the RISK and the SAFE pathway [73,74]. However, as the SAFE pathway in humans is under the control of STAT5, the role of STAT3 in this pathway is still a matter of debate [75]. 

### 3.3. Nitric Oxide and cGKI Pathway 

Cyclic guanosine-monophosphate (cGMP) and cGMP-dependent protein kinase type I (PKG aka cGKI) are known to prevent both I/R damage and cardiac remodeling [76]. The nitric oxide (NO)–sensitive guanylyl cyclase (NO-GC aka s-GC) is the most relevant target of cGKI [77,78,79,80]. Cardiac cGKI exerts its cardioprotective effects [81] by opening the mitoBKCa, which results in potassium influx [82] and mPTP closing. A direct effect of NO via S-nitrosilation of mitochondrial proteins (mitoSNO) has been also suggested to play a role in this pathway [83].

### 3.4. Autophagy

Autophagy is a self-phagocytic phenomenon in which lysosomes degrade intracellular molecules and organelles [84]. Autophagy is considered to be an adaptive and protective response of cardiomyocytes to ischemia [85] and it is involved in cardioprotection that is induced by remote ischemia preconditioning (rIPC) [86]. The inhibition of the mammalian target of rapamycin (mTOR), via adenosine monophosphate (AMP)-activated protein kinase (AMPK), is the main mechanism that is involved in autophagy [87]. Therefore, as expected, AKT-dependent mTOR activation prevents autophagy [for a review on mTOR refer to [88]]. However, a beclin-dependent autophagy has been shown to worsen tissue damage during reperfusion [85]. In particular, it has been reported that, while partial beclin inhibition exerts beneficial effects, its knockdown increases cell death [85,88,89,90].

## 4. Current Strategies to Reduce Ischemic Damage and Reperfusion Injury

The failure to develop new therapeutic options that are able to effectively prevent reperfusion injury fully reflects the complexity of this process. In the last decades, a number of different strategies have been investigated (Figure 1). Herein, the most relevant pharmacological, non-pharmacological, and interventional approaches will be discussed (Table 1).

### 4.1. Pharmacological Approaches

Historically, coronary artery disease (CAD) and myocardial ischemic damage were clearly defined by the concept that “time is muscle”. Back-to-back studies demonstrated that early reperfusion effectively protects myocardium from I/R damage. The demonstration that coronary thrombosis is the most relevant mechanism of damage [113] has spurred the development of drugs that are able to interfere with the burden of intracoronary thrombosis. Indeed, streptokinase was the first intracoronary approach that was exploited for the treatment of acute MI, in 1976 [114]. Nevertheless, despite improvement in managing CAD, it is becoming even more evident that unsolved issues should be taken on. Multitarget pharmacological approaches and promising mechanical reperfusion techniques have been developed to face this challenge.

#### 4.1.1. Current Multitarget Therapy: Antiplatelet Drugs and Beta-Blockers

Platelets, besides playing a crucial role in the first phase of thrombus formation, undergo activation during fibrinolysis. The efficacy of antiplatelet/thrombolytic combo treatment was first demonstrated by the ISIS-2 trial [91], proving advantageous in terms of mortality in the arm of aspirin plus streptokinase versus placebo or single drug. Afterwards, aspirin gained a central role in patients undergoing PPCI, as mechanical reperfusion progressively replaced thrombolysis [115]. Aspirin mainly acts by inhibiting the cyclooxygenase (COX). However, platelet activation and aggregation is under the control of thromboxane A2 during thrombin formation. Therefore, the impact of combining aspirin and drugs inhibiting the adenosine diphosphate receptor P2Y12, such as clopidogrel [92,116], prasugrel [93], and ticagrelor [94], was established.

The effects of several beta-blockers have been investigated in order to reduce oxygen consumption and decrease ischemic damage. However, only the intravenous administration of metoprolol before PPCI was able to significantly reduce the infarct size in patients with STEMI [117]. The inhibition of neutrophil-platelet interactions has been proposed to explain the metoprolol effect on reperfusion damage [118].

#### 4.1.2. Anti-Inflammatory Drugs

Given the pivotal role of inflammation in accelerating atherosclerosis and CAD, a number of preclinical and clinical studies have been performed. In particular, due to the crucial role of Interleukin-1 (IL-1) in mediating ischemia-induced inflammatory response, several IL-1 inhibitors have been investigated in preclinical and clinical studies. In rats, the anti-IL-1b antibody, gevokizumab, prevented HF progression [119]. Similarly, the administration of the IL-1 receptor antagonist, anakinra, in STEMI patients, improved the LV volume indices [96,120]. However, while treatment with anakinra was associated with a reduction of inflammatory markers, it caused a higher incidence of major adverse cardiovascular events (MACE) at 12 months, in 182 Non–STEMI (NSTEMI) patients [97]. Canakinumab administered every three months led to a lower rate of recurrent cardiovascular events and lung cancer; however, ahigher incidence of fatal infections was reported [98]. Data on potential combo-treatment are still missing.

#### 4.1.3. Adenosine and Sodium Nitrite

In pilot studies, adenosine administration was reported to reduce the infarct size [121,122]. The larger AMISTAD II trial confirmed this trend demonstrating that three hours adenosine infusion (70 μg/kg/min.) impacts on the infarct size. However, it failed to prove clinical benefits [99]. A post-hoc analysis in patients that were treated within 3.17 h from the onset of evolving anterior STEMI, a significant protection against early and late mortality was proved [100]. Unfortunately, recent trials analyzing the effects of high doses of adenosine administered intracoronary failed to demonstrate real benefits on myocardial damage and microvascular perfusion [101,102]. Similar to adenosine, the beneficial effects of Sodium Nitrite that were observed in MI preclinical studies were not validated in a randomized clinical trial enrolling 229 STEMI patients undergoing PPCI [103]. Therefore, the impact of adenosine and Sodium Nitrite are still a matter of debate. Further insight into the therapeutic efficacy of these pharmacological options could be potentially unraveled by future combo-treatments. 

### 4.2. Non-Pharmacological Approaches

#### 4.2.1. Ischemic Pre-Conditioning

The role of ischemic pre-conditioning was first described by Murry et al. [123]. They have shown that transient cycles of induced ischemia, followed by reperfusion substantially reduced the infarct size in dogs [123]. In addition, remote ischemic preconditioning (rIPC), consisting in four cycles of 5-min. brachial cuff inflations, resulted as effective as pre-conditioning performed during the occurrence of acute MI [104]. Unfortunately, the CONDI-2/ERIC-PPCI trial failed to demonstrate the improvement of clinical outcomes (cardiac death or hospitalization for HF at 12 months) of rIPC in STEMI patients undergoing PPCI [124]. The rIPC as a part of a multitarget therapy might be the future challenge. As a matter of fact, exenatide, in combination with rIPC (COMBAT-MI trial) (COMBinAtion Therapy in Myocardial Infarction trial) (NCT02404376), is ongoing and hopefully will provide new insight.

#### 4.2.2. Ischemic Post-Conditioning

In human studies, post-conditioning has been obtained by performing cycles of 1-min. inflation and 1-min. deflation of the angioplasty balloon, just after reperfusion by direct stenting. This approach was proved to reduce Creatine Kinase (CK), a surrogate marker of the infarct size, in a prospective, randomized, controlled, multicenter study [105]. Moreover, long-term benefits in LVEF recovery were reported [106]. Nevertheless, Cardiac Magnetic Resonance (CMR) failed to demonstrate clear-cut benefits in terms of outcomes and myocardial recovery in larger randomized trials [107,108]. Moreover, when evaluated in STEMI patients during PPCI, it failed to prove advantages in the composite outcomes, including death from any cause and hospitalization for HF [109]. Again, a combo-treatment involving rIPC and post-conditioning (CARIOCA trial: Combined Application of Remote and Intra-Coronary Ischemic Conditioning in Acute myocardial infarction) (NCT03155022) is ongoing, and the results will be provided in 2021. 

### 4.3. Interventional Strategies

Interventional approaches are spreading in cardiology, and new invasive therapeutic options have been investigated to avoid I/R injury. The reduction of myocardial oxygen consumption or an improvement of heart collateral blood supply were expected. 

#### 4.3.1. Left Ventricle Unloading

An improvement of myocardial salvage was reported using Intra-Aortic Balloon Pump (IABP) before reperfusion in the first preclinical studies [125,126]. Unfortunately, the Counterpulsation to Reduce Infarct Size Pre-PCI Acute Myocardial Infarction (CRISP AMI) trial [110] failed to demonstrate a reduction in the infarct size, in patients with anterior STEMI, without shock at presentation, and routinely undergoing to IABP before PPCI. However, a significant reduction of mortality in a subgroup of the CRISP AMI patients, with ST segment deviation >15 mm and persistent ischemia, has been reported [111]. 

Alternative approaches, such as the Impella^®^ heart pump, are under investigation. In preclinical studies, trans-valvular left ventricle unloading with Impella^®^ was found to limit MI and promote the expression of genes that are associated with mitochondrial respiration [127]. Evidence of feasibility and safety was provided by the Door-To-Unload in STEMI Pilot Trial (DTU-STEMI), involving patients with anterior STEMI without shock and randomized in two arms: patients with LV unloading with Impella^®^ followed by immediate reperfusion and patients that were subjected to 30 min. LV unloading before reperfusion [128]. Similarly, ECMO (extracorporeal membrane oxygenation) have also provided interesting insights in terms of cardiac protection, although its application is associated with a number of complications, thus should be limited to high risk setting [129].

#### 4.3.2. Pressure-Controlled Intermittent Coronary Sinus Occlusion (PICSO^®^) 

PICSO^®^ consists of a balloon-tipped catheter that is placed into the coronary sinus (CS). It has been shown that a balloon, alternately inflated and deflated, can intermittently increase the CS pressure and induce venous blood redistribution via collaterals. The high expression of vascular endothelial growth factor (VEGF) and hemoxigenase was found in experimental approaches applying PICSO^®^ [130]. Moreover, the reactivation of embryonic signaling pathways that are associated with both the induction of shear stress and blood flow pulsatile stretch have been proposed as the most relevant mechanisms that are associated to PICSO^®^-mediated beneficial effects [131]. Recently, the feasibility of PICSO^®^ in patients with ACS and positive physiological index of microvascular dysfunction was reported [112]. Further studies will provide new data.

## 5. EV and Cardioprotection

The lack of effective cardioprotective treatments has spurred both biologists and clinicians to move towards the progressive development of concepts for future therapeutic options. Cell-based therapies have drawn the path towards a new class of therapeutic strategies [132,133]. In the last decade, different stem cells, including mesenchymal stem cells (MSCs), adipose-derived stem cell (ADSCs), cardiac-derived progenitor cells (CPCs), embryonic stem cells (ESCs), cardiospheres-derived cells (CDCs), and induced pluripotent stem cells (iPSCs) have been proposed as cell-based therapy for cardiac repair after MI [132,134,135,136]. Cardiomyocyte (CM) proliferation, angiogenesis, and cardiac progenitor cell expansion have been deeply investigated [137,138]. However, increasing evidences have suggested that the beneficial effects that are derived from stem cell administration mainly relied on paracrine mechanisms that are also mediated by EV [139,140].

According to the Minimal Information for Studies of EV (MISEV) classification, three different EV subclasses have been identified. These subclasses include Exosomes (Exo), Microvesicles (MV), and Apoptotic Bodies [141].

Exo, the smallest EV, ranging from 30 to 100 nm, were first described by Pan and Johnstone [142,143]. Exo are generated and released as multivesicular bodies by a mechanism that is known as the endosomal sorting complex required for transport (ESCRT) [144,145,146,147]. A combination of several exosomal markers are commonly used for their characterization [148].

Mid-size EV range from 100 to 1000 nm and include microvesicles (MV), microparticles, or ectosomes. In this review, this EV subclass will be referred as MV. The most accepted mechanism of formation and release is the shedding after membrane budding [147]. Therefore, their cell of origin is defined by specific surface markers [149].

The largest EV, also known as apoptotic bodies, range from 800 to 5000 nm [147] and will not be discussed in this review.

In the last decades, EV have emerged as crucial mediators of biological signals among different cells to regulate discrete biological processes. As their action mainly recapitulates that of their cell of origin, a number of studies have been performed to investigate the healing properties of EV derived from different cell sources on scar formation and HF progression upon I/R damage [23]. When compared to stem cells, EV are theoretically more feasible, non-toxic, non-immunogenic, can be produced on a large scale, and can be adequately stored prior to their use.

## 6. MV and Cardioprotection

Circulating MV have been implicated in several physiological functions, such as the coagulation, reticulocyte maturation, angiogenesis, tissue repair, and inflammation [150,151,152]. In CAD, MV enriched in proinflammatory and procoagulant components have been mainly involved in the progression of atherosclerosis and the activation of coagulation [153,154,155]. Moreover, in ACS and atherosclerotic patients, the increased number of MV derived from platelets and endothelial cells (ECs) have suggested their potential use as disease biomarkers [14].

The role of MV in I/R damage is however controversial, since their effects could depend not only on the cell of origin, but also on the microenvironment of releasing cells. Herein, the protective or damaging effects exerted by MV derived from different cell types will be described (Table 2).

### 6.1. Platelet-Derived MV (PMV)

PMV were the first identified MV [13]. PMV are generally characterized by the expression of CD61 [163]. It has been shown that platelets release MV in response to several stimuli [164]. The Scott syndrome, which is an haemorrhagic disorder characterized by impaired MV formation, provided evidence for the relevance of PMV during coagulation [165]. PMV, by interacting with ECs, play a relevant role in the development and progression of atherosclerosis and vascular damage [155]. It has been reported that the elevation of PMV and EC-derived MV reflects the size of the injured myocardium during I/R, which suggests their possible application as biomarkers [166].

However, PMV can also play a beneficial role in I/R setting. Indeed, it has been reported that PMV locally injected induce angiogenesis and stimulate post-ischaemic revascularization in a rat model of MI. This protective effect relies on VEGF-mediated activation of the RISK Pathway [152].

In addition, PMV have been also involved in rIPC-mediated cardioprotection [156]. rIPC is able to increase the release of MV from platelet, ECs, erythrocyte, and leukocyte [163]. MV and PMV that are isolated after rIPC are able to reduce apoptosis in CMs [157], by inhibiting endoplasmic reticulum stress [158]. However, PMV that was isolated after rIPC failed to induce any protective effect in a different study [159]. Therefore, the role of PMV in rIPC is still debated.

### 6.2. Endothelial-Derived MV (EMV)

EMV, expressing CD144 or CD31, play an important role as markers of endothelial activation in several pathological conditions [167], and, as expected, their release is markedly increased during I/R damage [166]. However, they should be considered more than a simple marker of ischemia, as EMV released in this setting generate pro-apoptotic and pro-oxidative signals in CMs [160].

### 6.3. Other MV

MV can be released by different cells after I/R and can exert discrete actions. CM-derived MV after AMI (marked by the expression of Caveolin 3 and Troponin T) are internalized by infiltrating monocytes and regulate the local inflammatory response [161]. MV, released by MSCs overexpressing GATA-4, were found to be cardioprotective. This cardioprotective effect relies on MV enriched in miR-221, which reduces cell apoptosis by silencing the pro-apoptotic protein PUMA [162].

## 7. Exo and Cardioprotection

Exo that are derived from different stem cell sources are known to act by releasing their composite cargo, including lipids, proteins, and genetic information into recipient cells [168]. For cardiac regeneration, Exo miR cargo is the most extensively evaluated [21]. 

Although Exo can be released from different stem cells, it has been reported that Exo released by MSCs and CPCs are much more effective in term of cardioprotection and cardiac regeneration. Moreover, it has been demonstrated that stem cell-Exo exert their healing effects by a fine-tune control of processes involved in autophagy and inflammation. Therefore, cardioprotection, autophagy, and inflammation will be considered as showcases of Exo actions, and the MSC- and CPC-Exo properties will be much more deeply discussed. Finally, relevant data on the role of cardiac telocytes (CT) in cardioprotection will be briefly reported (Table 3).

### 7.1. Cardioprotection

#### 7.1.1. MSC-Exo

MSCs are nonhematopoietic multipotent stromal cells that are isolated from bone marrow able to differentiate towards mesodermal lineages [133]. Lim et al. [169] first demonstrated that cardioprotection induced by human MSC-conditionated medium (MSC-CM) and was mediated by Exo [200,201]. More recently, it has been shown that GATA-4 overexpression or ischemic preconditioning commit MSCs to release Exo able to prevent apoptosis, to reduce infarct size, and to improve cardiac function after MI [169,170,171]. The anti-apoptotic effect was attributed to miR-19a and miR-22 enriched in Exo [202]. Moreover, it has been reported that MSC-Exo mitigate oxidative stress and induce cardioprotection by activating the PI3K/AKT signalling pathway [203]. The results from a meta-analysis corroborated these data [204]. Among the miRs carried by Exo, miR-21 is one of the most relevant miR involved in cardioprotection [205]. In particular, it has been shown that miR-21 plays a crucial role in the activation of the RISK pathway by triggering AKT activation via PTEN downregulation [206,207,208,209]. As a matter of fact, MSC-Exo, enriched in miR-21 [172], activate AKT and GSK-3β [201,203] and inhibit mPTP opening-induced apoptosis.

miR-144 belongs to a cluster of miRs (miR-*144/451*) induced by GATA-4 [210]. Both miR-144 and miR-451 confer protection against in vitro I/R injury by targeting the COX-2 pathway [210]. miR-144 promotes cell survival through the phosphorylation of AKT, GSK-3β, and p44/42 MAPK [210,211]. miR-144 also attenuates cardiac I/R injury by targeting FOXO-1 [212], a protein that is involved in cardiomyocyte apoptosis. The finding that miR-144 was enriched in Exo recovered after rIPC and the loss of rIPC-mediated cardioprotection in miR144/451 knock-out mice support the crucial role of this miR cluster in rIPC-mediated cardioprotection [213].

#### 7.1.2. CPC-Exo

CPCs consist of a heterologous cell population resident in the adult heart, quiescent in physiological conditions, while being capable of undergoing differentiation into myocytes and vascular cells upon injury [214,215]. When cultured in suspension, CPCs form spherical aggregates, denoted as cardiospheres (CDCs) [216]. Similar to CPCs, CDCs release Exo displaying cardioprotective properties [217]. Indeed, it has been reported that the injection of both CPC- and CDC-Exo into the infarct border zone reduces the number of apoptotic CMs and prevents scar formation. CPC- and CDC-Exo enriched in miR-146 were also found to inhibit oxidant stress-induced cell death in rat CMs. Moreover, CPC-conditioned medium, containing Exo enriched in miR-210, reduce CM apoptosis via the downregulation of ephrin A3 and protein-tyrosine-phosphatase 1 (PTP1b), [182,183,184].

Recently, the presence of the pregnancy-associated plasma protein-A (PAPP-A), a protease that regulates the release of active insulin growth factor-1 (IGF-1), has been suggested to exert cardioprotective action upon CPC-Exo treatment [185]. Moreover, in rats that were exposed to I/R, treatment with CDC-Exo reduces macrophage infiltration and inhibits CM apoptosis [186]. This cardioprotective effect was confirmed in pigs that were subjected to I/R injury and treated with CDC-Exo [187]. The enrichment of miR-21 and miR-451 in CPC-Exo also reduced scar formation by inhibiting caspase 3/7-mediated apoptosis in CMs [188]. A relevant functional recovery was also observed in mice, treated with ESC-Exo [191]. Indeed, the intramyocardial delivery of ESC-Exo improves CM survival by inducing cyclin A2, D1, D2, E1 mRNA expression, and promoting neovessel formation at day 5 after MI [191].

### 7.2. Autophagy

Exo can trigger both activation or the inhibition of autophagy [218]. It has been reported that MSC-Exo promote cardioprotection during I/R by inducing autophagy through the AMPK pathway [173]. The enrichment of miR-30a in CM-derived Exo restrains beclin-pathway and autophagy, while the inhibition of miR-30 expression prolongs autophagy and cell survival in an in vitro model of I/R [195]. MSC-Exo enriched in miR-125b and ADSC-Exo enriched in miR-93-5p were found to reduce autophagy and improve cell survival [174,192]. The anti-apoptotic protein BCL-2 inhibits autophagy and actively participates in the cross-talk between autophagy and the RISK/SAFE pathway, by acting as a STAT3 downstream effector [87,175]. Indeed, BCL-2 is a crucial node in cardioprotection that is mediated by Exo derived from different cell types, including ADSCs, MSCs, and human umbilical cord-derived MSCs (huMSCs) [22,175,176]. miR-24 was found to be enriched in huMSC-Exo [219]. miR-24 controls the cardioprotective pathways, by modulating the expression of both Bim, and the autophagy-related gene 4a (ATG4A). Therefore, miR-24 has been proposed as a therapeutic target of I/R injury [220].

### 7.3. Modulation of Inflammation

As demonstrated by knocking-down CD4, the activation of CD4+ T-Helper has a beneficial effect in reducing LV dilation after MI [221]. Exo derived from Dendritic cells (DC-Exo), by enhancing CD4+ activation and guiding the inflammatory response toward the Th1 pathway, exert a beneficial effect on post-MI cardiac function [198]. Immune suppression was also obtained when B cells were subjected to cardiac-EC-Exo [222]. Moreover, miR-181a enriched MSC-Exo drives Treg differentiation and limits I/R damage [177]. Finally, it has been shown that β2-microglobulin knock-down in huMSCs, translates into the release of Exo more effectively than the wild types in inhibiting cardiac fibrosis. This effect also relies on the immune response modulation [219].

### 7.4. Cardiac Telocytes (CTs)-Exo 

Particular attention has been recently focused on the role of CT-Exo in cardioprotection [223]. Telocytes, are specific interstitial cells, which were identified in different organs and tissues [224]. Telocytes have been described in all cardiac layers and in the stem cell niches [225]. In the last ten years, their role in cardiac protection has been explored, as the cross-talk between CPCs and CTs contributes to cardiac regeneration [226]. In MI preclinical model, CT replacement significantly decreases the infarct size and improves cardiac function by inducing pro-angiogenic signals [196]. CTs promote angiogenesis by releasing Exo enriched in a number ofmiRs, including: miR-let-7e, miR-10a, miR-21, miR-27b, miR-100, miR-126-3p, miR-130a, miR-143, miR-155, and miR-503 [197,227].

## 8. Exo and Cardiac Regeneration

After damage tissue regeneration depends on the expansion of resident stem-progenitor cells, however, as compared to the fetal heart, a low self-renewal capability has been reported in the adult human heart [228]. Therefore, the most relevant challenge in cardiology would be the development of strategies that are able to rescue tissue damage after I/R injury by expanding cells or recovering their derivatives with regenerative properties. The identification of resident cardiac stem cells (CSCs) and the possibility to reprogram fibroblasts into CMs have opened a promising field of research [133,229,230]. However, transplanted in vitro-cultured CSCs poorly engraft. Moreover, the tumorigenic potential of transplanted CSCs represents an additional ethical hurdle to move toward their clinical application [231,232,233]. To solve these issues, novel strategies exploiting Exo to enhance CSC proliferation or to reprogram fibroblasts in vivo have been proposed [133]. Moreover, the role of Exo in promoting angiogenesis and cardiac regeneration will be discussed, as new vessel formation is instrumental for tissue regeneration [217].

### 8.1. Angiogenesis

#### 8.1.1. CPC-Exo

It has been reported that the intramyocardial injection of CPC-Exo increases vessel density, reduces scar size, and improves LVEF recovery in preclinical MI models. These effects have been associated with miR-132 Exo content, which promotes neovessel formation by regulating the expression of the RasGAP-p120 protein [184,187]. A different study demonstrated that miR-322 engineered CPC-Exo promote angiogenesis by activating the Nox-2 pathway in ECs both in vitro and in vivo [189]. Moreover, the in vivo injection of CXC Chemokine Receptor 4 (CXCR4) enriched Exo, as released by engineered CPCs, besides increasing Exo cardiac homing, boosts local angiogenesis [190]. CDC-Exo have been also shown to stimulate angiogenesis and to decrease programmed cell death. In addition, they promote CM proliferation, thus improving cardiac function and cell viability after MI. Enrichment of miR-146 and miR-22 in CDC-Exo mediates these effects [183].

#### 8.1.2. MSC-Exo

The pro-angiogenic effect of MSC-Exo has been linked to the enrichment in PDGF (platelet-derived growth factor), EGF (epidermal growth factor), FGF (fibroblast growth factor), NF-kB [234], and in the extracellular matrix metalloproteinase inducer (EMMPRIN), a key regulator of matrix metalloproteinase activities [235]. However, Exo proangiogenic action could be improved through engineering or preconditioning approaches. Indeed, Exo that are released by MSCs pre-treated with atorvastatin are much more active in inducing angiogenesis. This depends on Exo cargo enriched in lncRNA H19. H19 transferred to ECs and CMs acts as a precursor by releasing one of its exon, miR-675. miR-675 induces EC and CM survival by regulating VEGF and ICAM-1 expression [178,236]. CXCR4 engineered MSCs also release pro-angiogenic Exo. This effect mainly relies on the Exo mediated activation of the IGF-1α/PI3K/AKT pathway [179]. Electroporated MSC-Exo enriched in miR-132 were also found to be effective in promoting angiogenesis [180].

#### 8.1.3. ADSC-Exo

Adipose cells have been exploited as a source of MSCs and their role in promoting angiogenesis has been asserted in several studies [237]. ADSC-Exo have been shown to prevent apoptosis after MI and promote angiogenesis [176,192]. These effects could be boosted by enriching ADSC in miR-126 and miR-146a [193,194].

#### 8.1.4. Plasma-Exo

It has been reported that Exo derived from plasma of MI patients boost ECs proliferation, migration, and tube formation via miR-939-iNOS-NO-mediated pathway, both in vitro and in vivo. Such Exo have different cell of origin. It has been postulated that they can be released by CMs that were subjected to ischemic stress or from resident cardiac ECs [199]. Indeed, EC-Exo enriched in miR-214 were found to promote angiogenesis [238]. 

### 8.2. Cardioregeneration

As extensively reported, scar formation should be prevented and tissue damage restored to improve long-term patient’s outcomes. This implies that efforts should be directed to the identification of cells or much better their derivatives able to induce cardiac regeneration by expanding resident CSCs. A number of studies have been published. Herein, only data that unquestionably demonstrated the contribution of stem cell-Exo in promoting regeneration by expanding resident CSCs have been reported.

It has been shown that ESC-Exo promote cardiac repair after MI by supporting angiogenesis and increasing the survival and proliferation of c-kit+ CSC both in vitro and in vivo. miR-294, enriched in ESC-Exo, mediates CSC expansion [191].

MSC-Exo also improve proliferation of c-kit+ CSC in vitro and enhance their engraftment after transplantation. The improved engraftment depends on MSC-Exo expressing miR-760 and miR-326, which induce the angiogenic switch and myocyte differentiation [181].

## 9. Therapeutic Device by Manipulating Exo

Exo possess several therapeutic advantages when compared to stem cells. They are biocompatible, non-immunogenic, non-tumorigenic, and more stable in the circulation. Moreover, they cross the blood–brain barrier (BBB) [168]. The lipid layer of Exo provides protection from circulating enzymes, while the expression of surface protein provides an efficient homing and drug delivery to target cells, which minimizes potential side effects. However, several hurdles have to be solved before we can move to clinical application. Herein, the most relevant approaches for achieving this goal will be examined.

### 9.1. Exo Isolation and Production

The isolation method is still tricky. While various techniques have been evaluated, currently the most widely accepted is the differential ultracentrifugation. Differential ultracentrifugation provides high Exo purity, and is thereby optimal for research purposes. However, it requires special equipment and the low-yield still remains the main drawback for the transition to clinical application [141]. Additional approaches, such as microfluidic-based techniques [239,240,241], are under investigation to deal with this issue. However, further standardization is required [141].

The selection of robust sources of Exo is still a matter of debate for their production and scalability. As plasma is enriched in Exo (~10^10^/mL), and plasma-derived Exo have been reported to induce cardioprotection [139], plasma-Exo derived from healthy donors were first investigated [242]. However, risks of contamination and potential side effects have raised concerns in their clinical application [243]. Therefore, stem cells that were cultured in vitro are currently considered to be the safer and more manageable Exo source. As extensively reported, the Exo cell of origin dictates their specific effects. However, differences in the Exo yield have been reported when diverse cell sources have been cultured in vitro. High Exo production (~10^13^/mL) has been reported for CPCs [244] and MSCs [245]. This has provided the feasibility for manufacturing. Good Manufacturing Practices (GMP) for the production of therapeutic-oriented Exo have been therefore drawn up [for a complete review of the state-of-art of Exo manufacturing see [246]]. The immortalization of MSCs by c-myc transfection has been also used to obtain a stable cell line, enabling a scalable manufacturing procedure to therapeutic Exo mass-production [247]. However, the risk that is associated with the transfer of tumorigenic cues could raise ethical concerns.

### 9.2. Exo Targeting to Increase Cardiac Homing

Cell targeting is still a crucial issue, since the liver homing impairs Exo tissue distribution after intravenous injection [248]. Consistently, in pig subjected to MI, intramyocardial injection was found to be much more effective than intracoronary delivery in terms of microvascularization and scar formation [187]. However, the risk that is associated with intramyocardial injection should be prevented if translation to the clinic would be pursued. In addition, Exo are more promptly internalized by ECs and fibroblasts than CMs, hampering the possibility to directly induce CM proliferation [139]. Moreover, Exo have a fast clearance in vivo due to the lack of support of the extracellular environment and the scavenger action of macrophages [248,249]. Therefore, the boost of Exo survival after in vivo injection is still required. huMSC-Exo encapsulation in functional hydrogels, which mirror the presence of extracellular matrix proteins, is a novel strategy for increasing Exo stability and survival after in vivo injection. This approach was found effective in improving myocardial function [250].

New delivering approaches that are based on Exo surface modification have been proposed. Data on cardiac-homing peptide (CHP)-tagged Exo have provided promising results on cardiac homing to the infarct area, induction of CM proliferation, angiogenesis, and scar size [251]. A 15% increase of Exo delivery was also reported in both in vitro and in vivo experiments while using Exo expressing the cardiac-targeting peptide (CTP) bound to Lysosomal-associated membrane protein 2b (Lamp2b) [170,252]. The CXCR4 expression on CPC-Exo was found to enhance CM targeting and induce cardioprotection [190]. Finally, a significant increase in Exo delivery into the ischemic myocardium was obtained by tagging MSC-Exo with the CSTSMLKAC peptide sequence, also named ischemic myocardium-targeting peptide (IMTP) [170,253].

Novel approaches for increasing Exo targeting have been proposed in cancer. Drug-loaded Exo targeting cancer cells has been tested. The most intriguing approach so far provided is magnetic-field Exo targeting, as obtained by iron-oxide nanoparticle preloading [254,255]. Further studies are required to deeply investigate the feasibility of this approach in different clinical settings, including I/R.

### 9.3. “Drug” Loaded Exo

Exo contain many active components, including proteins and RNAs. Insight into the components that play a role in their regenerative capacity is still under investigation. Even if Exo could be considered a feasible therapeutic option by themselves, their functional capability could be optimised by modifying their cargo. Owing to their effect on epigenome, miRs are the most studied component of the Exo cargo, and the manipulation of their expression is the most promising approach in regenerative medicine. Indeed, many preclinical studies have been performed by upregulating the expression of specific miRs in Exo to induce cardiac regeneration. Ischemic preconditioning, either in vitro or in vivo, is an effective method for upregulating the expression of specific miRs in Exo [170,211,212,213]. Moreover, MSC preconditioning with statins was also found effective in enhancing cardioprotection and angiogenesis by changing Exo lnc-RNA and miR cargo [178].

More selective methods involve direct cell manipulation by genetic approaches. Knocking-down β2-microglobulin in MSC using the Clustered Regularly Interspaced Short Palindromic Repeats (CRISPR) strategy [219], or upregulating the expression of sonic hedgehog in human CD34+ (SHH) [256] have been shown to enhance Exo biological effects, by modifying their miR cargo. The transfection of stem cells in vitro with modified lentivirus or plasmids [177,179,194,257] is probably the most effective strategy to load selected miRs in Exo. Different Exo producing cells, including MSCs and ADSCs, have been modified to this purpose [193,194].

Cell overexpression of selected miRs to obtain Exo enrichment has been extensively exploited and investigated in preclinical models recapitulating different clinical settings [21,217]. More recently, the possibility to selectively transfer miRs in Exo has been provided. In particular, the identification of SYNCRIP, a protein that is involved in miR exosomal sorting process, has been described [258]. The authors have elegantly demonstrated that SYNCRIP binds miR by recognizing specific hEXO sequences in miR [259]. Further studies improving Exo-specific miR loading would be the challenge for biotechnological applications and therapeutic approaches in the future.

Alternatively, extracellular loading strategies have been proposed, as the EV electroporation [260]. miR-132, loaded in MSC-Exo by electroporation, has been successfully delivered both in vitro and in vivo. Moreover, its expression was associated with new vessel formation and the preservation of cardiac function after MI [180]. The same approach has been employed to load miR-322 in CPC-Exo [189]. Although of interest, all of these Exo engineering approaches raised concerns. Multiple miRs could be up- or downregulated in Exo upon ischemic, drug preconditioning, or genetic manipulation. Moreover, direct Exo electroporation might lead to the loss of native protective miRs or proteins impairing Exo effectiveness. These issues should be solved before Exo clinical application.

## 10. Conclusions

Current therapeutic approaches to prevent or reduce long-term complications in MI patients pose a heavy social and healthcare burden. Pharmacological and non-pharmacological approaches, alone or in combination, have provided advantages in preventing ischemia-induced damage and improving patient’s outcomes. Interventional approaches as new invasive therapeutic options have been investigated. In particular, trans-valvular left ventricle unloading with Impella^®^ provided promising results in a preclinical study [127] and evidence of feasibility and safety in the Door-To-Unload in STEMI Pilot Trial (DTU-STEMI) [128]. Likewise, the feasibility of PICSO^®^ in a subset of ACS patients has been reported [112]. Moreover, multitarget approaches have been proposed and the combo-treatment that involves rIPC and post-conditioning (CARIOCA trial: Combined Application of Remote and Intra-Coronary Ischemic Conditioning in Acute myocardial infarction) (NCT03155022) is ongoing and will provide new results in 2021. However, the management of microvascular damage and HF progression are still a clinical challenge. This has spurred researchers and clinicians to explore novel therapeutic approaches able to interfere with or prevent scar formation and HF progression. Alternative therapeutic options, including stem cell-based therapies, have been proposed to support tissue regeneration and ameliorate long-term complications [133]. However, particular attention has been devoted to their paracrine derivatives, such as Exo and MV, due to the lacking long-term stem cell engraftment. Their proangiogenic and cardioprotective properties have been extensively reported [21,132,133,217]. It has been suggested that Exo that are released from the heart after rIPC are required for cardioprotection, sustaining the relevance of mechanisms involving vesicular transfer in cardioprotection [157,211,261]. It has also been suggested that circulating Exo and MV, in humans and rats, are protective in a Langendorff-perfused rat heart [157]. Therefore, plasma-Exo/MV themselves appear to be necessary for endogenous cardioprotective mechanisms. However, ultimate insight into their functional role is mandatory for defining their mode of action.

A number of miRs that were carried by Exo derived from different cell sources has been reported to drive cardioprotection [11,12,192,207,257]. Moreover, the optimization of their protective role have been extensively investigated and different approaches proposed to efficiently modify their cargo [21,180,257]. Of note, although limited, specific RNA-binding- proteins and miR consensus sequences that were involved in Exo loading have been described [258]. Insight into the molecular mechanisms regulating this process may allow the production of Exo engineered with specific subsets of miRs able to modulate the expression of genes involved in cardioprotection in a tailored way. However, so far, the knowledge and the proof of concept for the most cost-efficient sorting of Exo-miR package to produce Exo with a specific cargo are still missing.

Moreover, Exo-based drug development would require optimization, including the identification of feasible cell sources for a large production of functional Exo, as well as standardisation in production. Indeed, the standardization of protocols for Exo production to achieve reproducibility, large scalability, quality control, and legislation are still an unmet need. Moreover, a GMP-compliant production of therapeutic Exo has to deal with several hurdles, even though some studies provide interesting perspectives [244,245,246]. Finally, while different clinical trials are currently ongoing to evaluate their potential application, the standardization of clinical protocols for different stem cell-derived Exo requires further improvements, including feasibility and safety. As multitarget therapies lie ahead for the treatment of several clinical settings, the possibility to combine knowledge and mode of action of new interventional therapeutic options and Exo would be the future challenge to generate ready-to-use pharmacological tools (Figure 1).

## Figures and Tables

**Figure 1 ijms-20-05024-f001:**
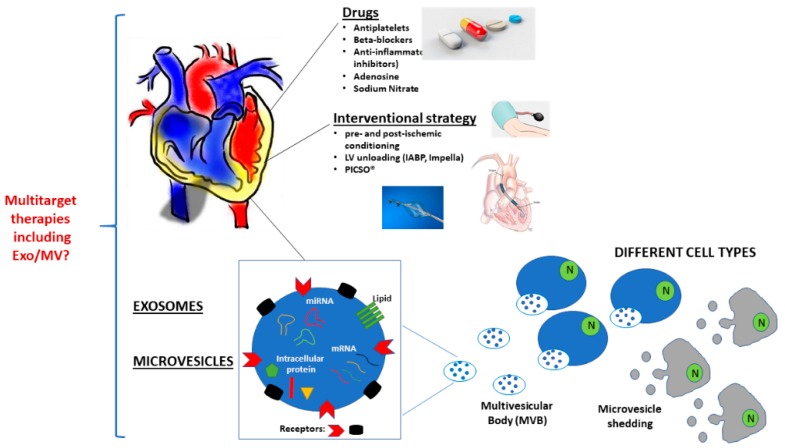
Current and future strategies to reduce ischemia/reperfusion (I/R) damage. Schematic representation of current pharmacological and interventional approaches to avoid long-term complication in MI patients are reported. In addition, a schematic representation of Exosomes (Exo) and Microvesicles (MV) is drawn. The possibility to exploit Exo and MV alone or in combination with pharmacological or interventional therapeutic options will represent the future challenge.

**Table 1 ijms-20-05024-t001:** Therapeutic strategies to reduce ischemic damage and reperfusion injury.

Study, Year	Population (N)	Design	Treatment	Primary Endpoints	Results	Refs
*ISIS 2, 1988*	Suspected acute MI (17187)	Multicenter, double-blinded, two-by-two factorial, placebo-controlled, randomized trial	Streptokinase vs. 1-month ASA vs. both vs. neither	Vascular mortality at 5 weeks, non-fatal reinfarction, bleeds requiring transfusion, non-fatal stroke, and cerebral hemorrhage.	The co-administration of streptokinase and ASA reduced vascular mortality compared to single drug treatment (40% vs. 23% vs. 20%).	[91]
*Cure, 2001*	UA/NSTEMI (12562)	Multicenter, double-blind, parallel group, placebo-controlled, randomized trial	ASA + clopidogrel vs. ASA + placebo	Composite of CV mortality, non-fatal MI, or stroke.	Dual antiplatelet therapy reduced CV mortality, non-fatal MI, or stroke but increased the rate of major bleeding	[92]
*TRITON-TIMI 38, 2007*	ACS (13608)	Multicenter, double-blind, randomized trial	ASA + prasugrel vs. ASA + clopidogrel	CV mortality, non-fatal MI, or non-fatal cerebrovascular events.	Prasugrel reduced CV morbidity and mortality but increases bleeding compared to clopidogrel	[93]
*PLATO, 2009*	ACS (18624)	Multicenter, double-blind, randomized trial	ASA + ticagrelor vs. ASA + clopidogrel	Vascular mortality, MI, or cerebrovascular events, major bleeding.	Ticagrelor reduced the rate of CV death, MI, or stroke without increasing the rate of overall major bleeding	[94]
*METOCARD-CNIC, 2014*	Anterior STEMI undergoing PCI (270)	Randomized trial	Metoprolol iv	Infarct size at 5–7 days (underpowered).	Beta-blocker was associated with a smaller infarct size compared with control; improved LVEF at 6 months	[95]
*VCU-ART, 2010*	STEMI (10)	Double-blind, placebo controlled, randomized trial	Anakinra vs. placebo	Change in LVESVi at CMR and echocardiography at 3 months.	Anakinra decreased LVESVi and LVEDVi	[96]
*MRC-ILA Heart Study, 2014*	NSTEMI (182)	Double-blind placebo-controlled, randomized trial	Anakinra vs. placebo	AUC for CRP over the first 7 days.	Anakinra reduced CRP levels, but increased the incidence of CV events at 12 months	[97]
*CANTOS, 2012*	Post-MI and elevated CRP (10061)	Double-blind, multi-center, placebo-controlled, randomized trial	Canakinumab (50, 150 or 300 mg)	Composite of nonfatal MI, nonfatal stroke, or cardiovascular death.	Canakinumab 150 mg reduced the composite outcome mainly reducing non-fatal MI; reduction in lung cancer, but associated with higher risk of fatal infections	[98]
*AMISTAD-II, 2005*	STEMI (2118)	Double-blind, multi-center, placebo-controlled, randomized trial	Adenosine infusion vs placebo for 3 h before PPCI/fibrinolysis.	New congestive heart failure beginning >24 h after randomization, or the first re-hospitalization for CHF, or death from any causes within six months.	No difference between placebo and adenosine. Adenosine dose-response relationship in decreasing median infarct size.	[99]
*AMISTAD-II post-hoc analysis, 2006*	STEMI (2118)	Double-blind, multi-center, placebo-controlled, randomized trial	Adenosine infusion vs placebo for 3 h before PCI/fibrinolysis.	New congestive HF beginning >24 h, or the first re-hospitalization for CHF, or death from any causes within six months. Endpoint analyzed according to time of reperfusion therapy.	Adenosine (<3.17 h) reduced mortality at both 1 and 6 months as well as the primary clinical endpoint at 6 months, with no distinction between adenosine dose regimens.	[100]
*David Garcia-Dorado et al., 2014*	STEMI (201)	Double-blind, placebo-controlled, randomized trial	Intracoronary infusion of 4.5 mg Adenosine vs saline immediately prior to reperfusion	Percentage of total myocardial necrotic mass assessed by CMR at 2–7 days post-reperfusion.	Intracoronary Adenosine administration prior to PCI did not limit infarct size.	[101]
*Desmet et al., 2011*	STEMI (112)	Prospective, double-blind, placebo-controlled clinical study	Intracoronary infusion 4 mg of Adenosine or matching placebo distal to the coronary occlusion site immediately before initial balloon inflation	Myocardial salvage defined as the percentage of the area at risk (AAR), which was not necrotic on CMR at day 2 and 3.	No evidence of changes in myocardial salvage.	[102]
*NIAMI, 2014*	STEMI (229)	Double-blind, multi-center, placebo-controlled, randomized trial	IV administration of 70 mmol sodium nitrite or matching placebo over 5 min immediately before PPCI	Difference in percentage of LV myocardial mass between active and placebo at 6–8 days post-infarct assessed by CMR.	No reduction in infarct size	[103]
*Bøtker* et al. *2010*	STEMI (251)	Prospective, single-center randomized controlled trial	rIPC (intermittent arm ischemia through four cycles of 5-min. of inflation and deflation of a blood-pressure cuff) vs nothing before PPCI.	Myocardial salvage index at day 30 after primary percutaneous coronary intervention, estimated by G-SPECT.	rIPC before hospital admission increases myocardial salvage.	[104]
*CONDI-2/ERIC-PPCI 2019*	STEMI (5401)	Single-blind, multi-center randomized controlled trial	rIPC (intermittent arm ischemia through four cycles of 5-min. of inflation and deflation of a blood-pressure cuff) vs. nothing before PPCI.	Cardiac death or hospitalisation for heart failure at 12 months	rIPC does not improve clinical outcomes	[104]
*Staat et al., 2005*	STEMI (30)	Prospective, multi-center, randomized, open-label, con- trolled study	Post-conditioning after PPCI performed within 1 min of reflow by 4 cycles of 1 min. inflation and deflation of the angioplasty balloon	Infarct assessed by measuring total creatinine kinase release over 72 h.	Post-conditioning reduced infarct size.	[105]
*Thibault et al., 2008*	STEMI (38)	Prospective randomized controlled trial	Post-conditioning after PPCI performed within 1 min of reflow by 4 cycles of 1 min. inflation and deflation of the angioplasty balloon.	Persistent infarct size reduction, assessed by SPECT imaging with rest-redistribution index at 6 months.	Post-conditioning affords persistent infarct size reduction	[106]
*POST, 2013*	STEMI (700)	Multi-center, randomized, open-label, blinded trial	Post-conditioning after PPCI performed within 1 min of reflow by 4 cycles of 1 min. inflation and deflation of the angioplasty balloon.	Complete ST-segment resolution (percentage resolution of ST-segment elevation >70%) measured at 30 min after PCI	Post-conditioning did not improve myocardial reperfusion in STEMI patients	[107]
*POST substudy, 2015*	STEMI (111)	Multi-center, randomized, open-label, blinded trial	Post-conditioning after PPCI performed within 1 min of reflow by 4 cycles of 1 min. inflation and deflation of the angioplasty balloon.	Myocardial salvage measured by CMR at day 3 after the index event.	Myocardial salvage index was not improved.	[108]
*DANAMI-3–iPOST, 2017*	STEMI (1234)	Multi-center, randomized clinical trial	Conventional PPCI vs post-conditioning performed as 4 cycles of 30-s balloon occlusions and reperfusion after opening of the infarct-related artery and before stent implantation.	A combination of all-causes of death and hospitalization for heart failure at follow-up.	Post-conditioning during PPCI failed to improve clinical outcomes.	[109]
*CRISP-AMI, 2011*	STEMI (337)	Multi-center, randomized clinical trial	Initiation of IABP before PPCI and continuation for at least 12 h (IABP plus PPCI) vs PPCI alone.	Infarct size expressed as a percentage of LV mass measured by CMR 3 to 5 days after PPCI.	IABP plus PPCI compared with PCI alone did not result in reduced infarct size.	[110]
*CRISP-AMI substudy, 2015*	STEMI (36)	Retrospective analysis	PPCI + IABP vs PPCI alone in large myocardial infarction and poor ST segment resolution	All-causes of mortality at six months, and composite endpoint of death, cardiogenic shock and new or worsening HF at six months.	IABP associated with decreased six-month mortality in large STEMI complicated by persistent ischemia after PPCI	[111]
*OxAMI-PICSO, 2018*	STEMI (105)	Single-center, investigator-initiated study, prospective study	PICSO in patients with index of microcirculatory resistance >40 compared to historical cohort of controls.	Infarct size assessment within 48 h after PPCI and at six months.	IMR-guided treatment with PICSO may be associated with reduced infarct size	[112]

ACS = acute coronary syndrome; AUC = area under the curve; CMR= Cardiac Magnetic Resonance; CRP = C-reactive protein; CV = cardiovascular; G-SPECT: by gated single photon emission CT; IABP = intra-aortic balloon pump; IV= intravenous; LVEF = left ventricular ejection fraction; MI = myocardial infarction; NSTEMI = non-ST segment elevation myocardial infarction; PPCI= primary percutaneous coronary intervention; rIPC= remote ischemic pre-conditioning; UA = unstable angina.

**Table 2 ijms-20-05024-t002:** MV role in ischemic myocardium.

Source	Animal Model	Administration	Effects		Mechanisms	Refs
			In Vitro	In Vivo		
Platelet	Rat MI	Intramyocardial	Angiogenesis EC proliferation	Angiogenesis	VEGF-PDGF—bFGF RISK pathway	[152]
Platelet from rat undergoing rIPC	Rat I/R	Intravenous	-	Improved cardiac function	Increase of MV circulating in periferial blood. Undefined	[156]
EV from coronary blood after rIPC	Langendorf—mode isolated heart	Intracoronary	-	Decrease of infarct size	Undefined	[157]
MV isolated after IPC in periferial rat blood	Rat LAD ligation	Intravenous	-	Decrease of infarct size Reduced cardiomyocyte apoptosis	Decrease of caspase-3 and -12 activity Reduced endoplasmic reticulum stress	[158]
MV isolated after IRC in periferial rat blood	Rat coronary ligation	Intravenous		Failure to decrease infarct size compared to MI alone without MV		[159]
MV isolated from HUVEC after H/R	H9c2 cardiomyocytes	Incubation in vitro	Increased apoptosis	-	Higher level of ROS and lipid peroxidation Bcl-2 inhibition	[160]
Cardiac MV isolated from cardiac ischemic tissue	Rat coronary ligation	Incubation in vitro of MV with Ly6+ monocyte	Modulation of inlammation	.	Increased release of Il6 and CCL2 and CCL7	[161]
MV collected from MSC overxpressing GATA-4	Cardiomyocyte after H/R	Incubation in vitro	Reduced apoptosis	-	miR-221 overexpression modulation of PUMA	[162]

EC = endothelial cells; EV= extracellular vesicles; HUVEC = Human umbilical vein endothelial cell; H/R = hypoxia/reoxigeniation; IPC = ischemic preconditioning; IRC = ischemic remote conditioning I/R = ischemia/reperfusion; LAD = left anterior descending artery; MI = myocardial infarction; MiR = microRNA; MSC = mesenchymal stem cell; MV= microvesicles; PUMA = p53 upregulated modulator of apoptosis; rIPC= remote ischemic pre-conditioning; RISK = Reperfusion Injury Salvage Kinase.

**Table 3 ijms-20-05024-t003:** Exo in cardioprotection/cardioregeneration.

Donor Cells	Animal Model	Administration	Effects		Mechanisms	Refs
			In Vitro	In Vivo		
Mesenchymal stem cell (MSC)						
MSC-conditioned medium (MSC-CM)	Mouse I/R	Intravenous	Undefined	Reduction of infarct size	Undefined	[169]
MSCs following ischemic preconditioning (EXOIPC)	C57BL/6J mouse LAD ligation	Intramyocardial	Anti-apoptosis	Reduction of cardiac fibrosis	miR-22 targets methyl CpG binding protein 2 (Mecp2)	[170]
MSC overexpressing GATA-4 (ExoGATA-4)	Mouse LAD ligation	Intramyocardial	Increase of CM survival, reduction of CM apoptosis and preservation of mitochondrial membrane potential	Recovery of contractile function reduction of the infarct size	Anti-apoptotic miRs (e.g., miR-19a), by reducing PTEN expression drive the activation of the Akt-ERK signalling pathway	[171]
Endometrium-derived mesenchymal stem cells (EnMSCs)	Mouse MI	Intramyocardial	Anti-apoptosis Angiogenesis	Anti-apoptotic effects Angiogenesis myocardial salvage and improvement of cardiac function	mir-21, PTEN, Akt pathway	[172]
MSCs	Rat I/R	Intramyocardial	Anti-apoptosis Autophagy	Increase sautophagy, reduction of apoptosis and myocardial infarct size	AMPK/mTOR and Akt/mTOR pathway	[173]
Transplanted MSCs	Mouse MI	Transplantation	Autophagy reduction	Autophagy reduction	miR-125b modulates p53-Bnip3 signalling	[174]
Human-derived MSCs	Isolated rat heart I/R	Intramyocardial	Autophagy and apoptosis inhibition	Cardiac function recovery	BCL2 up-regulation	[175]
Adipose-derived MSCs (ADMSCs)	Mouse I/R	Intramyocardial	Anti-apoptosis	Reduction of infarct size	Wnt/β-catenin signaling pathway	[176]
B2M deletion-human Umbilical Cord Mesenchymal Stem Cells (B2M-UMSC)	Rat MI	Intramyocardial	Undefined	Cardiac fibrosis inhibition, cardiac function recovery	mir-24/Bim pathway	[22]
MSCs	Mouse I/R	Intramyocardial	Anti-inflammation Treg polarization	Anti-inflammation, delayed ischemic damage	mir-181a (lentiviruses), c-Fos inhibition	[177]
Atorvastatine-pretreated MSCs (MSCATV-Exo)	Mouse MI	Intramyocardial	Angiogenesis Anti-apoptosis	Cardiac function improvement, infarct size reduction, anti-apoptotic effects, angiogenesis and anti-inflammation	lncRNA H19 regulation of miR-675, activation of VEGF and ICAM-1	[178]
MSC transduced with lentiviral CXCR4	Rat MI	Intramyocardial	Anti-apoptosis Angiogenesis	Angiogenesis, infarct size reduction, improvement of cardiac remodelling	IGF-1α and pAkt up-regulation, active caspase 3 downregulation, VEGF enhancement	[179]
MSCs	Mouse LAD ligation	Intramyocardial	Angiogenesis	Angiogenesis, heart function preservation	miR-132, RASA1 gene	[180]
Cardiac stem cells (CSCs) preconditioned with MSC-EXO	Mouse LAD ligation	Intramyocardial	Proliferation, migration, and tube formation of c-kit+ CSCs	Angiogenesis, reduction of fibrosis, LV function recovery	Upregulation of miR-147, let-7i-3p, miR-503-5p, and miR-362-3p	[181]
Cardiac-derived progenitor cell (CPC)						
CPCs	Mouse MI/R	Intramyocardial	Anti-H_2_O_2_ induced apoptosis	Anti-apoptotic effects	miR-451	[182]
CDC-conditioned medium (CDC-CM)	Mouse MI	Intramyocardial	Angiogenesis, anti-apoptotic effects and proliferation	Reduction of the scar mass, improvement of cardiac function	miR-146a, suppression of Irak1 and Traf6 (TLR pathway), NOX-4 and SMAD4 (TGF-β pathway)	[183]
Human derived-CPCs	Mouse LAD ligation	Intramyocardial	Anti-apoptotic effects and angiogenesis	Reduction of the scar mass, angiogenesis, improvement of cardiac function	miR-210 -> down-regulation of ephrin A3 and PTP1b miR-132 -> down-regulation of RasGAP-p120 miR-146a-3p	[184]
Human derived CPCs and bone marrow-derived mesenchymal stem/progenitor cells (BMCs)	Rat MI and I/R	Intramyocardial	Anti-apoptotic effects (CPCs > BMCs)	Reduction of the scar size, improvement of LVEF (CPCs> BMCs) in I/R model (CPCs only) Angiogenesis	PAPP-A (Exo-CPC), IGF-1 release, activation of the Akt-ERK signaling pathway	[185]
CDCs	Mouse I/R	Intracoronary	Protection against oxidative stress	Reduction of infarct size	Y RNA fragment (EV-YF1) induces IL-10 secretion	[186]
Human-derived CDCs	Pig Acute and chronic MI	Intracoronary Intramyocardial	Undefined	Reduction of infarct size (Acute MI) Reduction of the scar size (Chronic MI)	Alteration of pro-inflammatory and pro-fibrotic pathway	[187]
Mouse-derived CPCs	Mouse CMs Oxidative stress	Undefined	Anti-apoptotic effects	Undefined	miR-21 downregulates PDCD4, inhibition of caspase 3/7-mediated apoptosis	[188]
Mouse-derived CPCs	Mouse LAD ligation	Intravenous	Angiogenesis	Angiogenesis Infarct size reduction	miR-322 (transfection), Nox2-dependent H_2_O_2_ production	[189]
CXCR4-overexpressing CPC (ExoCXCR4)	Rat I/R	Intravenous	Anti-apoptotic effects	Infarct size reduction, LV function improvement	Increased cardiac homing	[190]
Embryonic Stem Cell (ESC)						
Mouse-derived ESCs	Mouse MI	Intramyocardial	CPC survival, proliferation, and cardiac commitment	Neovascularization, cardiomyocyte survival, reduction of fibrosis. CPC survival, proliferation, and cardiac commitment	miR-294, induced expression of cyclins (E1, A2, and D1)	[191]
Adipose-derived stem cell (ADSC)						
ADSCs	Mouse MI	Intramyocardial	Reduction of autophagy, apoptosis and inflammatory response	Reduction of autophagy	miR-93-5p-mediated suppression of hypoxia-induced autophagy and inflammatory cytokine expression by targeting Atg7 and Toll-like receptor 4 (TLR4)	[192]
miR-146a-modified ADSCs	Mouse LAD ligation	Intravenous	Anti-apoptotic anti-inflammatory, and anti-fibrotic effects	Anti-apoptotic, anti-inflammatory, and anti-fibrotic effects	Downregulation of EGR1	[193]
miR-126-overexpressing ADSCs	Mouse LAD ligation	Intravenous	Anti-inflammatory, anti-fibrotic, angiogenesis	Reduction of infarct size and cardiac fibrosis, angiogenesis	Spred1, PI3KR2/VEGF signalling pathway	[194]
Cardiomyocyte (CM)						
AMI patients CMs	H9C2 cardiomyoblasts	Undefined	Autophagy	Undefined	Inhibition of miR-30a or release of Exo increased expression of the core autophagy regulators beclin-1, Atg12, and LC3II/LC3I	[195]
Cardiac telocyte (CT)						
Mouse-derived CTs	Mouse LAD ligation	Intramyocardial	Undefined	Infarct size reduction, Cardiac function improvement Angiogenesis	CTs and endothelial cell contact VEGF and NOS2 secretion Various miRNA	[196,197]
Dendritic cell (DC)						
Murine cultured bone marrow derived DCs (BMDCs)	Mouse MI	Intravenous	Activation of CD4(+) T cells	Improvement of cardiac function	Increased expression of chemokines and cytokines (IFN-γ and TNF)	[198]
Plasma						
Human coronary serum from ischemic patients	Mouse Limb ischemia	Intramuscular	Endothelial cell proliferation, migration and tube formation	Angiogenesis	miR-939-iNOS-NO pathway	[199]

ADMSCs = adipose-derived MSCs; ADSC = adipose-derived stem cell; AMI = acute myocardial infarction; B2M-UMSC = B2M deletion-human Umbilical Cord Mesenchymal Stem Cells; BMCs = bone marrow-derived mesenchymal stem/progenitor cells; BMDCs = murine cultured bone marrow derived DCs; CDC = cardiosphere-derived cell; CDC-CM = CDC-conditioned medium; CM = cardiomyocyte; CPC = cardiac-derived progenitor cell; CSCs = cardiac stem cells; CT = cardiac telocyte; DC = dendritic cell; EnMSCs = endometrium-derived mesenchymal stem cells; ESC = embryonic Stem Cell; I/R = ischemia/reperfusion; LAD = left anterior descending artery; LV = left ventricle; MI = myocardial infarction; MiR = microRNA; MSC = mesenchymal stem cell; MSCATV-Exo = atorvastatine-pretreated MSCs; MSC-CM = MSC-conditioned medium.

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
