# Peer review of "Ischemia Reperfusion Injury: Mechanisms of Damage/Protection and Novel Strategies for Cardiac Recovery/Regeneration"

_ijms, 2019, doi:10.3390/ijms20205024_

Round 1

Reviewer 1 Report

In the manuscript entitled “Novel Strategies for Cardiac Protection and Regeneration.” Caccioppo et al. have summarized the mechanism of cardiac damage and regeneration post-myocardial infarction with clinical applications of exosomes in cardioprotection. Although the review article represents well-summarized literature on the regenerative and cardioprotective properties of exosomes, there are some major concerns which need to be addressed.

Concerns:

The subject reviewed in this manuscript is likely of interest to the readership of the journal. However, with all due respect to the authors, the narrative is confusing and does not justify with the title of the manuscript. Almost half of the review text focuses on the pathological mechanisms and current treatment options, which are all well-reviewed in various reviews. Authors should include more information related to novel therapeutic interventions than existing interventions and the mechanism of cardiac damage. There are some studies which already explained the role of different cell-derived exosomes in cardiac regeneration post-myocardial infarction and reperfusion injury. [1-4] It is not clear what new information is provided in this current review. Authors have primarily focused on the exosomes, and the microparticles are almost neglected. In the introduction about novel therapeutic strategies for cardiac regeneration post-MI potential of microvesicles should also be acknowledged, and literature regarding microparticles in cardiac protection be included.

References:

Li N, et. al. New Insights into the Role of Exosomes in the Heart After Myocardial Infarction. J Cardiovasc Transl Res. 2019;12(1):18-27. Xu JY, et. al. Exosomes: A Rising Star in Falling Hearts. Front Physiol. 2017; 8:494. Davidson SM, Yellon DM. Exosomes and cardioprotection - A critical analysis. Mol Aspects Med. 2018; 60:104-114. Zhao W, et. al. Exosome and its roles in cardiovascular diseases. Heart Fail Rev. 2015; 20(3):337-48.

Author Response

 ANSWER TO REVIEWER 1

We thank the Reviewer for his/her evaluation and comments.

The subject reviewed in this manuscript is likely of interest to the readership of the journal. However, with all due respect to the authors, the narrative is confusing and does not justify with the title of the manuscript. Almost half of the review text focuses on the pathological mechanisms and current treatment options, which are all well-reviewed in various reviews. Authors should include more information related to novel therapeutic interventions than existing interventions and the mechanism of cardiac damage. There are some studies which already explained the role of different cell-derived exosomes in cardiac regeneration post-myocardial infarction and reperfusion injury. [1-4] It is not clear what new information is provided in this current review. Authors have primarily focused on the exosomes, and the microparticles are almost neglected. In the introduction about novel therapeutic strategies for cardiac regeneration post-MI potential of microvesicles should also be acknowledged, and literature regarding microparticles in cardiac protection be included.

References:

Li N, et. al. New Insights into the Role of Exosomes in the Heart After Myocardial Infarction. J Cardiovasc Transl Res. 2019;12(1):18-27. Xu JY, et. al. Exosomes: A Rising Star in Falling Hearts. Front Physiol. 2017; 8:494. Davidson SM, Yellon DM. Exosomes and cardioprotection - A critical analysis. Mol Aspects Med. 2018; 60:104-114. Zhao W, et. al. Exosome and its roles in cardiovascular diseases. Heart Fail Rev. 2015; 20(3):337-48.

To address the Reviewer’s comments the Ms has been extensively revised.

In the present version of the Ms the major point raised by the Reviewer has been clarified in the Abstract section and  in the last paragraph of the Introduction section as follows:

 Abstract“  …. this review will first summarize mechanisms of cardiac damage and protection after I/R damage to track the paths through which more appropriate interventional and/or molecular-based targeted therapies should be addressed. Moreover, it will provide insights on novel non-invasive/invasive interventional strategies and on Exo-based therapies as a challenge to improve patient’s long-term complications. Finally, approaches to improve Exo healing properties, and topics still unsolved to move towards Exo clinical application will be discussed.

Introduction “Since the therapeutic strategies are mainly based on specific targeting approaches, the first part of this review will introduce the most relevant mechanisms of damage and cardioprotection. Moreover, it will provide an overview on multitarget pharmacological/non-pharmacological approaches and on the promising mechanical reperfusion techniques developed to protect myocardium from I/R damage. In addition, recent advances at using EV for cardioprotection/cardiac regeneration will be reported. Finally, approaches to improve EV healing properties, and the hurdles still unsolved to move from bench to bedside will be discussed.

Introducing the mechanisms of I/R injury is crucial for a widespread reader to move to interventional and molecular-based therapeutic approaches we have described. Moreover, as pointed by the Reviewer, a number of reviews on this topic have been published. We have tried to arrange the Ms, as better as we can, in order to avoid duplication. We mainly focused on novel interventional and non-interventional approaches which, alone or in combination, are under investigation in clinic. Furthermore, we focus on the latest ongoing trials exploring combo-treatment options. Finally, we have described hurdles still existing for the use of Exo in clinic, without going in deep to avoid duplication.

To introduce the readers to the multitarget therapies, which is considered the future challenge, only a brief introduction on pharmacological approaches, which are currently available in clinical settings and were proved as effective, were reported. Since a review on multitarget therapies has been recently published (ref 4), we simply report the current available results on this topic.

As kindly suggested by the Reviewer microparticles have been included in the present version of the Ms. Likewise, for the indicated references (refs 22, 24,25,26). A new Table corresponding to Table 2 has been included.

The title has been changed accordingly.

Reviewer 2 Report

 The present review described several mechanisms of cardiac damage and protection after myocardial infarction and provided comprehensive insights about therapeutic approaches for cardiac protection and regeneration, especially focusing on exosomes. The current version scientifically follows the topic of exosomes clinical application and touches related hurdles and intriguing perspectives.  It’s qualified to the acceptance in International Journal of Molecular Sciences, but minor misspelling and the ragged space between characters are found.

Author Response

ANSWER TO REVIEWER 2

We thank the Reviewer for his/her appreciation and positive comments on the review.

The present review described several mechanisms of cardiac damage and protection after myocardial infarction and provided comprehensive insights about therapeutic approaches for cardiac protection and regeneration, especially focusing on exosomes. The current version scientifically follows the topic of exosomes clinical application and touches related hurdles and intriguing perspectives.  It’s qualified to the acceptance in International Journal of Molecular Sciences, but minor misspelling and the ragged space between characters are found.

As kindly suggested by the Reviewer, the misspelled words and the ragged space between characters have been corrected.

Round 2

Reviewer 1 Report

The authors have addressed all my concerns. Authors have now included a section on the role of microvesicles in cardioprotection with an appropriate change in manuscript title with revised abstract section.